# Ecological Effect of Differently Treated Wooden Materials on Microalgal Biofilm Formation in the Grado Lagoon (Northern Adriatic Sea)

**DOI:** 10.3390/microorganisms11092196

**Published:** 2023-08-30

**Authors:** Vanessa Natali, Francesca Malfatti, Tamara Cibic

**Affiliations:** 1Oceanography Section, National Institute of Oceanography and Applied Geophysics-OGS, 34151 Trieste, Italy; vnatali@ogs.it; 2Department of Life Sciences, University of Trieste, 34127 Trieste, Italy; fmalfatti@units.it

**Keywords:** biofilm, microalgae, biofouling, wood treatment, impregnating agent, lagoon

## Abstract

Within the framework of the Interreg Italy–Slovenia programme, the project DuraSoft aimed at testing innovative technologies to improve the durability of traditional wooden structures in socio-ecologically sensitive environments. We focused on the impact of different wood treatments (i.e., copper-based coatings and thermal modification) on microbial biofilm formation in the Grado Lagoon. Wooden samples were placed in 2 areas with diverse hydrodynamic conditions and retrieved after 6, 20, and 40 days. Light, confocal and scanning electron microscopy were employed to assess the treatment effects on the microalgal community abundance and composition. Lower hydrodynamics accelerated the colonisation, leading to higher algal biofilm abundances, regardless of the treatment. The Cu-based agents induced modifications to the microalgal community, leading to lower densities, small-sized diatoms and frequent deformities (e.g., bent apices, frustule malformation) in the genera *Cylindrotheca* and *Cocconeis*. After 20 days, taxa forming 3D mucilaginous structures, such as *Licmophora* and *Synedra*, were present on chemically treated panels compared to natural ones. While in the short term, the treatments were effective as antifouling agents, in the long term, neither the copper-based coatings nor the thermal modification successfully slowed down the biofouling colonisation, likely due to the stimulating effect of nutrients and other substances released from these solutions. The need to develop more ecosystem friendly technologies to preserve wooden structures remains urgent.

## 1. Introduction

In the Italian lagoons, the use of wood for infrastructure such as piers, moorings, pilings and fences is part of the local heritage [1]. However, these structures are subjected to constant maintenance that is no longer sustainable in humid and coastal environments where wood degradation is massive and fast [2]. To overcome this problem, in the last century, very toxic wood coating products have been used, causing environmental contamination [3]. Within the framework of the Interreg Italy–Slovenia project DuraSoft (innovative technologies to improve the durability of traditional wooden structures in socio-ecologically sensitive environments), we have tested some new coating techniques and products developed with the aim of increasing the durability of traditional wood species. The environmental compatibility of these techniques has been evaluated through a broad spectrum of ecotoxicological tests [3]. One of the project objectives was also to evaluate the ecological effect of differently treated wooden materials on the microalgal biofilm formation in the Grado–Marano lagoonal system. In the present study, the effort was focused on the microalgal components of this aquatic ecosystem that are at the base of the trophic web but are usually neglected in ecological surveys.

Microbial biofilm, or microfouling, forms on all kinds of natural and artificial substrates immersed in marine or lagoon environments. It is composed of different microorganisms, of which the major components are microbes and diatoms, and it is surrounded by a matrix of extrapolymeric substances (EPS, [4]). These mucilaginous substances protect the biofilm against chemical stressors [5,6], by absorbing various molecules and ions but also heavy metals and other contaminants. Molecules absorbed into EPS may not reach the cells, and because of this, biofilms can grow in polluted environments where other organisms often cannot [7,8]. Biofilms play an important role in mediating other essential functions and biogeochemical processes, such as nutrient enrichment and hydraulic stress [9]; however, at the same time, they are especially vulnerable to physical disturbance [10,11]. Moreover, a mature stage of biofilm favours the engraftment of animal organisms, or macrofouling, which represents one of the major problems of structures immersed in the marine environment since its presence can eventually lead to the degradation of all submerged surfaces, regardless of the material [12,13]. To the best of our knowledge, there is no literature focused specifically on the effects of wood impregnating products on the development and structure of marine microalgal biofilms, although there is a large body of literature on antifouling paints used to delay biofouling formation on ship hulls [14,15,16]. In the past, macrofouling was controlled by painting ship hulls and man-made structures with toxic compounds such as tributyltin (TBT). At present, in most parts of the world, TBT and its derivatives can no longer be used due to their higher polluting power and strong impacts on benthic and planktonic organisms [17]. After the banning of TBT in the late 1980s in France, and from 2003 in the rest of Europe, Cu-based antifouling paints have progressively substituted for TBT-based paints [18] and are still being used nowadays. Indeed, copper-based biocides belong to the most important active ingredients in wood preservatives in Europe. The main reason for their widespread use is the good ratio between efficacy and toxicity, and the fact that most of the competing products have been banned. Therefore, copper is still allowed to be used in all use classes, including use class 5 (sea water applications) [19], whereas, to meet legislative requirements, chromium compounds in wood preservatives were replaced with amines, predominately ethanolamine [20].

Biocidal coatings remain the most popular choice to solve biofouling issues and still dominate the market, reportedly accounting for more than 90% of coating sales [21,22], although concerns over the potential environmental impact of biocides have led to increased attention being paid to the development of biocide-free approaches to fouling control [21]. The coating industry has an increasing interest in the development of biocide-free microfouling control solutions that rely on surface physical–chemical properties, although the development of a successful marine coating that is simultaneously effective against biofouling while being substantially environmentally benign is very challenging [13]. In the wood industry, an alternative to using impregnation solutions is represented by wood thermal modification, i.e., a controlled pyrolysis process of wood being heated (>180 °C) in the absence of oxygen to induce some physical–chemical changes in the structures of the cell wall components (i.e., lignin, cellulose and hemicellulose) in order to increase its durability [23]. It is therefore interesting to test these technologies in the field.

Coastal lagoon ecosystems are very sensitive environments, and they provide a variety of ecosystem services [24]. Thus, they were classified as priority habitats in the European Union Habitats Directive (92/43/EEC) and in the last European Water Framework Directive (WFD, 2000/60/EC) under the category “transitional water”. Notwithstanding, antifouling products are commonly being used in lagoons to reduce the presence of aquatic organisms that attach to boats or other structures. Antifouling coatings that leach heavy metals, predominantly copper or tin, have been effective in preventing colonisation by most fouling organisms (e.g., [25]); however, these coatings have also been found to cause major environmental impacts [26]. Therefore, it is of paramount importance to develop new formulations of wood antifouling impregnating agents, with lower Cu concentrations, that are still effective against biofouling but lead to lower leaching of contaminants into the environment. In parallel, the effectiveness of other treatments based on physical processes, such as wood thermal modification, have to be tested in situ, especially during the first stages of biofilm formation.

Biofilms are difficult to study and control, and much remains to be learned about their role in aquatic ecosystems [27]. Nevertheless, to control the formation of microfouling and improve the performance of antifouling coatings, it is important to understand the complex interactions between the biofilm and its environment. This includes studying the influence of environmental factors on the formation and structure of the EPS matrix, as well as on the microbial and microalgal community composition of the biofilm [8]. By understanding these interactions and dynamics, it is possible to develop strategies to reduce the formation of microfouling and to improve the performance of antifouling agents [28].

Here, we investigated the effect of differently treated wooden materials on the microalgal fraction of the biofilm. We evaluated the toxic effects of diverse treatments in terms of changes in the microalgal abundance, community composition and diatom cell damage by applying several microscopy techniques (i.e., inverted light microscopy for quantitative estimation, laser confocal scanning for qualitative estimation, scanning electron microscopy to observe cell damage). We compared a new formulation of the copper treatment (Silvanolin, Silvaprodukt, Ljubljana, Slovenia^®^) that should be more environmentally friendly with another antifouling impregnating agent commonly present on the market and another physical treatment, such as thermal modification. Furthermore, we evaluated how diverse hydrodynamic conditions influence the development of microalgal biofilm.

## 2. Material and Methods

### 2.1. Study Area

The Grado–Marano lagoonal system is located between the Tagliamento and Isonzo River mouths in the northern Adriatic Sea and is divided into two basins: Marano, characterised by a shallow water body and significative freshwater inputs, and Grado, a shallower (<1 m, on average) basin with several islands, salt marshes and with a complex network of canals [29]. With an area of approximately 131 km^2^, this lagoon is the second largest transitional water body along the Italian coast [30]. The exchange with the open sea (Gulf of Trieste) occurs through six inlets: Lignano, S. Andrea, Porto Buso, Morgo, Grado and Primero. The Natissa River is the only river that flows into the eastern part of the lagoon (near Grado) and, therefore, this area has decidedly more marine characteristics than the western part of the lagoon (Marano) [31]. The particulate matter inputs from small rivers are of secondary importance, since the estuaries of the Tagliamento and Isonzo Rivers are the primary source of sediments (silty and clayey particles, respectively) conferred to the lagoon [29].

The Grado–Marano lagoonal system is one of the best conserved wetlands in the whole Mediterranean area [29]. This system is classified as a coastal microtidal lagoon of large dimensions (Italian Decree n. 131/08), and it has been protected by the Ramsar Convention since 1971. Following the implementation of the Habitats Directive (92/43/EC), the lagoon was also designated as a Site of Community Importance (SCI-IT3320037). It is subjected to several stressors, both natural (tides, storms, floods) and anthropogenic (contamination, development, land use, etc.). In particular, the lagoon’s water quality has been deteriorating due to the increasing presence of nutrients and several contaminants as well as the discharge of untreated sewage, agricultural and industrial wastes. In addition, the lagoon is affected by the impacts of urbanisation, fishing activities, fish and clam farming and tourism, such as the construction of beach infrastructure, the destruction of wetlands, and the increase in boat traffic [32]. These factors are causing changes in the water quality and biodiversity of the lagoon, which in turn are having a negative impact on the socio-economic activities in the area [33].

The lagoon is strongly influenced by semi-diurnal tides, with a mean range of 0.90 m [30], that act as a forcing factor for the dissolved oxygen content since these tides greatly affect the lagoon hydrodynamics [34]. The confined area in the easternmost part of the Grado basin is characterised by scarce water exchange and the strong modification of the hydrological regime that occurred as a consequence of a bridge built in 1936 that connects the urban area of Grado to the inland area [34].

Based on current velocities [30], two different sites in the Grado basin were selected to run the experiment: “Schiusa” area, an area with higher hydrodynamics (20 cm/s), and “Approdo” area, an area with lower hydrodynamics (10 cm/s) (Figure 1).

### 2.2. Experimental Design and Sampling

Wooden panels of *Abies alba* (fir) and *Picea abies* (spruce) species measuring 7.6 cm × 2.6 cm × 0.5 cm were prepared by the project partner Silvaprodukt d.o.o. (Ljubljana, Slovenia): 48 samples were impregnated with a new-generation formulation of Silvanolin (24 samples with a copper concentration of 1% and 24 samples with a copper concentration of 0.25%); 48 samples with a product present on the market based on chromium, copper and boron salts (CCB, 24 samples with a copper concentration of 1% and 24 samples with a copper concentration of 0.25%); 24 samples were thermally modified (see details below) and 24 were prepared without any protective treatment to be used as reference samples. The wooden specimens were designed to be tested in certain experimental situations in accordance with the “Use Classes” (UCs) contained in the EN 335 standard [19]. For the impregnation process, the samples were vacuum-pressure impregnated (Silvanolin and CCB). Impregnation was carried out according to the full-cell procedure (30 min vacuum at 10 kPa; 3 h pressure at 900 kPa; 15 min vacuum at 20 kPa). The samples were subjected to a conditioning procedure over a month at a temperature of 20 °C and 70% relative humidity. The commercial solution of Silvanolin UC5 consisted of 4.34 g/kg of copper hydroxide carbonate, 2.43 g/kg of boric acid, 2.0 g/kg of quaternary ammonium compound, benzyl-C12-16-alkyldimethyl chloride and 0.025 g/kg of N-(3-aminopropyl)-N-dodecyl-propane-1,3-diamine [35]. The ethanolamine solutions at different concentrations of copper (cCu) were for UC3: cCu 0.25% and for UC5: cCu 1%.

For the thermal modification, the samples were prepared according to the Silvapro^®^ commercial procedure [36], i.e., isothermally treated for 3 hours at 190, 200, 210, 220 and 230 °C, respectively.

All of the wooden samples, divided and labelled according to the area, the experimental time, the type of wood and the type of treatment, were mounted on special structures called collectors (Appendix A). A total of 24 collectors were set up: the first 12 were positioned in the “Schiusa” area, while the remaining 12 were positioned in the “Approdo” area (Appendix A). Only samples obtained from the same typology of treatment were mounted on one collector to avoid contamination among different treatments, i.e., a collector was dedicated only to the Silvanolin treatments; therefore, on it, 8 panels were positioned:Two replicates of fir wood panels treated with Silvanolin 1%.Two replicates of spruce wood panels treated with Silvanolin 1%.Two replicates of fir wood panels treated with Silvanolin 0.25%.Two replicates of spruce wood panels treated with Silvanolin 0.25%.

The collector dedicated to the CCB treatments was similarly composed, while a collector with 4 replicates (2 fir + 2 spruce) of thermally modified wood panels and a collector with 4 replicates (2 fir + 2 spruce) of natural wood panels were also prepared (Appendix A).

The experiment was performed between 9th June and 19th July 2021. All of the collectors were kept in the water at the same depth (50 cm from the water surface, regardless of the tidal range) using a system of buoys. The retrieval of the collectors took place at the following experimental times:T1 (June 15): after 6 days.T2 (June 29): after 20 days.T3 (July 19): after 40 days.

The experimental times had to be kept short because, in the summer period, when the experiment was conducted, macrofouling quickly took over, especially in the “Approdo” area subjected to low hydrodynamics.

In total, 144 wooden samples were retrieved, 72 for each area, 48 for each experimental time, and stored in 50 mL Falcon tubes with 20 mL filtered (0.22 μm) seawater and pre-filtered and neutralised formaldehyde (final concentration 4%) buffered solution with CaMg(CO_3_)_2_. Once in the laboratory, they were rinsed with deionised water and stored at −20 °C in the dark with aluminium.

The analyses of the microalgal biofilm on the wooden panels were carried out to evaluate the possible toxic effects in terms of changes in the abundance, community composition and cell damage using several microscopy techniques. Due to the fast development of macrofouling on the wooden panels, the biofilm on samples retrieved at T2 from the “Approdo” area and at T3 from both areas could not be analysed. On these, the macrofoulers were quantitatively and qualitatively analysed, although the results are not included in the present paper but only briefly mentioned in the discussion.

### 2.3. Abundance and Community Structure of Microalgal Biofilm via Light Microscopy

The quantitative estimation of the microalgal community was obtained using light microscopy. A fraction of the attached biofilm (2 cm^2^, about 1/10 of the entire area of the wooden panel) was mechanically removed from each wooden panel by scraping with a scalpel. Then, it was transferred to a 15 mL vial with 2.5 mL of pre-filtered (0.2 μm) seawater and formaldehyde (final concentration of 4%) buffered solution with CaMg(CO_3_)_2_. The obtained aliquot was sonicated 3 times for 1 minute, with a 30 s pause in-between. Subsequently, 2.5 mL of this aliquot was placed into a counting chamber and the microalgae were counted and identified. If the microalgae were too abundant to count, a fraction of the initial aliquot (2.5 mL) was diluted in filtered (0.22 μm) seawater from 1:2 to 1:21, depending on the sample. Only cells containing pigments and not empty frustules were counted under a Leitz inverted light microscope (Leica Microsystems AG, Wetzlar, Germany) using a 32× or 40× objective (320× or 400× final magnification) [37]. The formula applied to obtain the number of cells per cm^2^ was the following:Abundance (cm^2^) = N * V_F_/2 V_C_ whereN = number of counted cells;V_F_ = final sample volume (mL), depending on the applied dilution;V_C_ = volume of the counting chamber (mL);2 = division factor to express the abundance from 2 cm^2^ to cm^2^.

The qualitative identification of the microalgae was carried out using the [38,39,40,41,42,43,44,45,46,47] identification keys, whereas the taxonomy was based on the Algaebase [48] and WoRMS [49] websites [50].

### 2.4. Microalgal Biofilm via Confocal Microscopy and Electron Microscopy

The qualitative estimation was obtained using laser confocal scanning microscopy (LCSM, NIKON C1si TE-2000U–Confocal Microscope System). Small pieces of wood of different shapes and sizes were prepared, taking care to remove only the surface part of each wooden panel. Then, they were immersed in the immersion oil for the observations.

The autofluorescence spectra of the pieces of wood were recorded with the following laser settings: 405 nm, 488 nm and 561 nm, and the laser powers were kept at around 30%. We used a 20× Plan Apo objective. We also tried the 40× and 60×, but the roughness of the wooden panel did not allow us to find the focus on the specimens. At the time of the analysis, the red laser 640 nm better suited for chlorophyll excitation was undergoing repair. We used NIS-Elements C version 5.0 and NIS-Elements Viewer software version 4.60 to perform the Z stack and volume rendering image analysis.

The qualitative cellular damage analysis was conducted via a scanning electron microscope equipped with an environmental analysis module (E-SEM FEI Quanta 250). With this microscope, it was possible to observe the sample in the hydrated state, without modifications, as with the usual processes (dehydration and metallisation) required by a normal SEM. The samples were prepared by cutting about a quarter of each wooden panel and directly mounting it on a stub. The samples were observed with a focused beam of electrons, with energy between 25 and 30 keV, at 400–800× magnification to obtain an overview of the biofilm on the wood piece but up to 1600–3000× magnification to observe the cell damage.

### 2.5. Statistical Analyses

A preliminary test of normality, the Shapiro–Wilk test, was performed to verify if the microalgal abundance data had a normal distribution. Since it deviated significantly from normality (*p* value < 0.001), an independent-samples T-test (Mann–Whitney U test) was carried out to evaluate whether there were differences in the abundances between the two types of wood. Furthermore, other pair tests were carried out to evaluate if there were significant differences between the treatments in both areas at the first sampling time (T1). All of these tests were carried out through JASP [51].

All further statistical analyses were performed using PRIMER 7.0.21 [52]. A matrix was created with the microalgal abundances considering the entire floristic list. Before the multivariate analysis, the biotic matrix was square-root transformed. A cluster analysis was performed by applying the triangular Bray–Curtis similarity matrix and complete linkage. To visualise the differences in the taxa assemblages among the different sampling areas, sampling times and treatments, a non-metric multidimensional scaling ordination (nMDS) [53] was performed on the Bray–Curtis similarity matrix. To highlight which taxa mainly contributed to the temporal and spatial variation in the assemblages, the taxa with the highest (average ≥ 5%) relative abundance (RA) were overlaid on the nMDS plot. Furthermore, the relative contribution of each taxon to the average similarities between the treatments of the wooden panels was calculated using a one-way similarity percentage procedure (SIMPER, cut-off percentage: 70%).

## 3. Results

The photographic time course of the organism colonisation of the wooden panels is shown in Appendix A, where it is possible to macroscopically observe the gradual colonisation of organisms from the biofilm phase after 6 days to the more structured fouling phase, which took place over 40 days. After 20 days, a greater colonisation of animal fouling was observed in the area with lower hydrodynamics, i.e., in “Approdo” than in “Schiusa”, where all of the wooden panels displayed a much smaller area covered by macrofoulers, regardless of the treatment.

### 3.1. Abundance and Community Structure of Microalgal Biofilm

The Mann–Whitney U test revealed there were no significant differences in the microalgal total abundances between the two types of wood (*Abies alba* and *Picea abies*) (U test; *p* value: 0.963); therefore, they were considered as replicates in further analyses. The total microalgal abundance estimates obtained via light microscopy evidenced different inhibiting effects among the treatments and between the two areas (Figure 2).

Considering both areas together and only the first sampling time (T1), we obtained a statistically significant difference between the Silv 1%, Silv 0.25% and CCB 0.25% treatments and the natural panel (Mann–Whitney U test; *p* value: 0.029 for all pairs of tests; Appendix A).

At T1, in “Schiusa”, the inhibitory effect of the impregnating agents was evident, as revealed by the lower cell abundances on the wooden panels treated with CCB 1% and CCB 0.25% (on average, 86 ± 17 and 308 ± 91.9 cells/cm^2^, respectively; Figure 2a). In addition, the wooden panels treated with Silv 1% and Silv 0.25% (on average, 652 ± 65.1 and 517 ± 9.9 cells/cm^2^) showed an inhibitory effect on the microalgal development compared to the natural one (on average, 5119 ± 458.9 cells/ cm^2^). This was also supported by the images taken via the confocal microscope (LCSM) and scanning electron microscope (SEM). In fact, very few cells were detected on the surface of the samples treated with Silvanolin and CCB compared to the natural sample, on which a higher cell density and larger cells were found (Figure 3).

In contrast, at T1 in “Approdo”, higher abundances were observed on all of the wooden panels compared to “Schiusa” (Figure 2a,b). The microalgal abundance on the thermally treated panels was comparable to that observed on the natural ones (13,142 ± 2457.9 and 8574 ± 5623.6 cells/cm^2^, respectively). Compared to the latter two, the panels treated with the impregnating agents displayed lower abundances, as can be deduced from the LSCM and SEM images (Figure 4), which also revealed mainly very small cells on the samples treated with chemical solutions (Figure 4d). Furthermore, some diatom cells, especially *Cylindrotheca* and *Cocconeis*, on the chemically and thermally treated panels showed frequent cellular damage: bent apices or other malformations of the siliceous diatom frustule that were detected under both the optical microscope and SEM (Figure 4a). The abundance values on the panels treated with Silv 1% were up to 33 times lower than those estimated on the natural wooden panels (on average, 8574 ± 5623.6 cells/cm^2^), while those treated with CCB 0.25% were up to 25 times lower than the reference panels. In addition, it is interesting to note that on the *Abies alba* panel with CCB 1%, the cell density was particularly high compared to the other treatments (9120 cells/cm^2^), anticipating the pattern observed in “Schiusa” at T2 (Figure 2b).

In “Schiusa” at T2, a completely different situation was noted compared to T1, i.e., higher cell densities were found on the panels treated with the impregnating agents, in particular on the *Abies alba* treated with CCB 1% and Silv 0.25% (30,940 and 24,760 cells/cm^2^, respectively; Figure 2c). Interestingly, the cell densities in the biofilm developed on the panels treated with Silv 1% were comparable to those on the natural wooden panels, which, among all of the investigated treatments, displayed the lowest abundance (6730 ± 5557.9 and 4672 ± 4113.2 cells/cm^2^, respectively).

By analysing the composition of the community at the genus level, we identified 18 genera within the Bacillariophyceae class, accounting for 99.63% of the whole microalgal community. Three genera were identified within the Cyanobacteria (relative abundance, RA = 0.35%) *Anabaena*, *Oscillatoria* and *Spirulina*. We were able to identify only one genus of Dinophyta, namely *Prorocentrum*, with very low RA (0.02%). Therefore, for further comparisons, we focused only on the diatom community and the differences in the relative abundance (RA) of the main genera developed on the samples (Figure 5). Unfortunately, in all of the samples, a fraction of the community consisted of very small (<10 µm) pennate diatoms, not possible to identify under the light microscope, and therefore we grouped them as “undetermined Pennales”.

In the “Approdo” area, at T1, *Cylindrotheca* was the dominant diatom genus on the natural and thermally treated panels (RA = 58.2% ± 16.68 and 58.6% ± 3.61, respectively), while it displayed lower RAs in the treatments with impregnating agents, especially compared to Silv 1% and Silv 0.25% (RA = 10.3% ± 4.67% and 25.2% ± 19.9%, respectively; Figure 5). In contrast, other genera seemed tolerant, i.e., *Nitzschia*, which increased its cell density on both panels treated with Silv 1% but also on those with CCB 1%. Similarly, we observed higher abundances of the genus *Amphora* in the treatments (on average, RA = 7.4% ± 3.27%) than on the natural wood (RA = 4.2% ± 0.07%). Finally, some diatoms were not present in the treatments with impregnating agents, such as the genus *Achnanthes*.

In “Schiusa”, the genus *Navicula*, prevalent on the natural wood (RA = 17.5% ± 1.41%), maintained a high RA even in the treatments with CCB and Silvanolin at both concentrations. In general, *Cylindrotheca* displayed lower densities on the natural panels (RA = 15.4% ± 0.52%) than in “Approdo”. The RA of this taxon increased in the treatment with Silv 0.25%, while it slightly decreased in the CCB treatments, with highly comparable data between the two CCB treatments. As already observed in “Approdo”, also in “Schiusa” the genus *Amphora* increased its abundance in the treatment with Silv 1% and CCB 0.25%, but above all, in the treatment with CCB 1% (RA = 11.3% ± 4.60%, 10.1% ± 1.51 and 13.6% ± 3.99%, respectively). At this site, specimens of *Achnanthes* were present on all of the panels, regardless of the treatment.

At T2, investigated exclusively in “Schiusa”, where the panels were not yet colonised by animal macrofouling, a modification of the structure of the microalgal community was noted on the treated panels: the first colonisers (e.g., *Cocconeis*, *Cylindrotheca*) greatly decreased or completely disappeared, while the genera adhering on the substrate by means of an adhesive disc or a mucilaginous peduncle took over (e.g., *Licmophora*, *Synedra*, especially on panels with Silv1%, but also *Fragilaria* and *Achnanthes*). In confirmation of this, these diatoms, particularly *Licmophora* and *Synedra*, were not much represented on the natural (RA = 0% and 3.4% ± 1.90%, respectively) or thermally modified panels (RA = 1.2% ± 2.69% and 3.9% ± 0.34%, respectively), while they became dominant on the panels treated with the impregnating agents (Figure 5).

### 3.2. Multivariate Analyses of Microalgal Biofilm

On the non-metric MDS ordination plot, calculated considering the total abundances of the microalgal community, we overlaid the most abundant taxa (RA ≥ 5%) and the six groups of samples based on the similarity higher than 60% (Figure 6). In the nMDS ordination, clear spatial differences among the areas and sampling times are evident: on the left side of the plot are placed samples within the “Schiusa” and “Approdo” areas at T1 with the lowest abundances, while on the rightmost part of the plot are those with the highest abundances, i.e., all of the samples from the “Schiusa” area at T2 (Figure 6). Furthermore, the treatment CCB 1% was separated from the others, likely based on its lowest abundance values compared to the other treatments in “Schiusa”. One group was constituted by four treated panels of “Schiusa” and two treated panels of “Approdo”, where the genus *Cylindrotheca* emerged as the discriminating factor. In the other groups, which comprised all of the samples from “Schiusa” at T1 and T2, the following genera were the most discriminating: *Synedra*, *Nitzschia*, *Achnanthes*, *Navicula* and *Amphora* (Figure 6).

Furthermore, the SIMPER analysis evidenced the highest dissimilarity between the Silvanolin 1% and CCB 1% treatments (57.44%), mainly due to higher abundances of *Cylindrotheca* sp. (contribution % = 11.11) and *Navicula* spp. (contribution % = 9.99) in the treatment with CCB 1%. The lowest dissimilarities were found between the thermo and natural panels (40.65%), where *Cylindrotheca* sp. (contribution % = 17.71) and *Synedra* spp. (contribution % = 7.68) were the most abundant taxa, particularly on the thermo panels.

## 4. Discussion

To test the antifouling properties of the Cu-based coating and the thermal modification, we chose to perform the experiment in the worst-case scenario, i.e., in the summer period (June–July) when the water temperature and light irradiance are the highest, thus enhancing the microalgal growth and substrate colonisation. These are the optimal conditions for biofouling proliferation in which to test the effectiveness of the antifouling agents. Our experiment provides new insights into the ecological effect of long-term Cu exposure on marine diatoms within microalgal biofilms. Considering only the first two experimental times, the Cu exposure lasted 20 days, which is much longer than the generation time of these microorganisms, as diatoms divide in a time range from hours to a few days [54]. Therefore, the applied exposure time could be considered long term or chronic, representing as much as possible an ecologically realistic setting that allowed ecological succession and competition to shape the communities under long-term chronic Cu exposure [18]. In fact, while laboratory studies mainly exclude the invasion of other species with higher tolerances to the selected contaminants, at the same time, they may bias the findings towards the synergistic effects of several stressors. This further demonstrates the importance of in situ studies mimicking the natural situation as close as possible [55].

The different current intensities in the two investigated areas strongly influenced the biofilm development. In the area with lower hydrodynamics, the colonisation process was accelerated, reaching microalgal abundances up to one order of magnitude higher than in the “Schiusa” area after only 6 days. This phase was then followed by the quick establishment of animal fouling, regardless of the applied treatment. In aquatic ecosystems, increased hydrodynamic shear forces were found to lead to an adaptation of the community; certain species likely better adapt to these conditions and thrive while others vanish [55]. In addition, in the marine realm, the velocity of the prevailing current was reported to be one of the main drivers of the development of different microphytobenthic assemblages in the Gulf of Trieste (northern Adriatic Sea) [56]. The authors found that in areas characterised by relatively weak bottom currents, mostly taxa that are loosely associated with the sediments proliferated. In contrast, in areas where the bottom currents were more intense, mostly species able to attach to the sediment grains were selected. This is in line with our findings, since in “Schiusa” the genus *Cylindrotheca* displayed much lower densities on the natural samples compared to the non-treated samples from “Approdo”. Indeed, in high hydrodynamic conditions, the tychopelagic (i.e., loosely associated with the substrate [57] diatom species *Cylindrotheca closterium* tends to be swept away by the current, whereas in mild hydrodynamic conditions, it develops reaching high numbers [58].

Antifouling agents such as those based on copper are commonly applied, although their toxic effects on the environment need to be carefully considered [16]. Bacillariophyta (diatoms) are known to be highly tolerant of Cu, which is linked with their capacity to synthetise extracellular polysaccharides [59] that chelate the Cu ion. The EPS matrix absorbs contaminants and protects the biofilms against chemical stressors [55]. However, higher Cu concentrations may induce several detrimental effects, both at the organism and community levels. In fact, although Cu is an essential element for microalgae [60], at higher concentrations it becomes toxic [15] by inhibiting CO_2_ fixation and PSII activity in photosynthetically active cells, causing oxidative stress and eventually inhibiting cell growth [59]. In our study, the first noteworthy effect of the tested impregnant solutions was an overall lower diatom density after six days, as consistently observed in both areas. In the samples treated with Cu-based agents, we observed small-sized diatom cells and frequent frustule deformities (bent apices, malformation of the frustule, deformed raphe), particularly in the genera *Cylindrotheca* and *Cocconeis*. According to Martinez and co-workers, the scarceness of specimens, lower dimensions and higher frequency of deformed valves are all responses to metal contamination [61]. Indeed, the authors reported high relative abundances of deformed diatom valves in contaminated sediments, mainly of *Achnanthes* spp. (up to 19%) but also of other genera such as *Cocconeis*, *Diploneis* and *Navicula*. They proposed *Achnanthes longipes* as a reliable reference to the response of benthic diatoms to metal contamination, branding this taxon a tolerating species. This is in accordance with our results, since we observed *Achnanthes* in high densities also on the panels with impregnating agents. Furthermore, we detected diatom cells belonging to the genera *Navicula*, *Nitzschia* and *Amphora* both on the treated and untreated samples. In “Approdo”, the area with lower hydrodynamics, the abundance of the genus *Amphora* increased in the treatment with CCB 0.25%, but particularly with CCB 1%, confirming its tolerance to heavy metals. Besides being metal-tolerant, *Amphora coffeaeformis* is one of the most common pioneer colonisers used in many studies as a model organism for underwater bio-adhesion [62]. However, *Amphora*, together with *Navicula* and *Nitzschia*, are the most widely distributed diatom genera in both contaminated and undisturbed ecosystems [61], thus having a high ecological valence and demonstrating high adaptability to all kinds of environments [63,64,65]. Within the same genus, some species may display a higher tolerance to a particular contaminant than others. During chronic exposure, Cu exerts species selection pressure on the biofilm community: sensitive species disappear and tolerant species dominate the community, resulting in an increase in the overall community tolerance [18,55]. Detailed descriptions of Cu-sensitive and -tolerant taxa in environmental communities are scarce and mostly report low values of diversity; yet, a reliable model recording the response of benthic diatoms to metal pollution is still lacking [61].

The short-term inhibitory effect (6 days) of both chemical impregnating agents was more effective in the area with higher hydrodynamics, although their effectiveness in inhibiting the biofilm development decreased after 20 days, when none of the tested treatments appeared to effectively counteract colonisation. This could be due to: (1) the dilution of heavy metals by sea currents, which favoured the removal of excess heavy metals, thus decreasing its inhibiting power; or (2) the release of quaternary ammonium compounds, present in Silvanolin, and the degradation of other organic substances, added to these impregnants, into nitrogen compounds, which likely promoted the proliferation of microalgae. Only at higher concentrations of Silvanolin 1% did the levels of heavy metals contained in the specimens presumably counteract the stimulating effect of the inorganic nutrients released from the panels. Interestingly, the cell densities in the biofilm developed on the Silvanolin 1%-treated samples were comparable to those on the natural wood specimens, which, of all the investigated treatments, presented the lowest microalgal abundance.

After 20 days of immersion in the area with higher hydrodynamics, we observed that the first diatom colonisers of hard substrates, such as the genera *Cocconeis* and *Cylindrotheca*, greatly decreased in numbers or even disappeared from the panels. In contrast, the applied chemical treatments selected diatom taxa, such as *Licmophora*, *Synedra* and *Fragilaria*, that adhere to the substate by producing adhesive mucilaginous stalks and tubes and form very complex three-dimensional structures (e.g., fan-shaped or star-shaped colonies) that are typical of a mature stage of biofilm. This typology of attachment to the substrate has a double advantage: (1) the cells can get closer to the source of light, towering over the adnate species that adhere to the substrate along the entire length of the cell, and, at the same time, (2) they move away from the source of contamination and reduce the toxic effects of the impregnating agents. In fact, the EPS matrix in which the cells are embedded absorbs herbicides and protects the biofilms against chemical stressors [6]. To confirm this hypothesis, these diatoms, especially *Synedra* and *Licmophora*, were not much represented on the natural or thermally modified panels compared to the chemically treated ones.

Alterations in the microalgal communities induced by antifouling agents have consequences for the further process of animal colonisation and, consequently, for the functioning of ecologically sensitive areas such as lagoons. As deduced from the photographic time course of the organism colonisation (Appendix A), after 20 days, none of the tested treatments prevented or drastically reduced the macrofoulers’ attachment. In the area with lower hydrodynamics, after 20 days, the abundance of macrofoulers on the specimens treated with both impregnants was much higher than that estimated on the control, probably due to the stimulation effect exerted by nutrients on the microalgal biofilm needed for the subsequent attachment of encrusting animals. Only in the Silvanolin 1% treatment was the presence of foulers comparable to that of the thermally modified sample. In contrast, in higher hydrodynamic conditions, after 20 days, only the thermal treatment proved effective in slowing down animal colonisation. A very different situation was observed after 40 days in the same area: the thermal treatment favoured the development and density of animal fouling the most. Silvanolin 1%, which retained its inhibiting efficacy best after 20 days, was the treatment with the highest animal density but the lowest number of species after 40 days, while the CCB treatments showed fouler abundance comparable to that of the natural panels.

When developing new techniques and antifouling products to increase the durability of wood structures used in lagoonal systems, one has to bear in mind that the new impregnating solutions or physical treatments have to be effective but, at the same time, environmentally friendly. There must be a trade-off between their antifouling effect and a low environmental impact due to the release of contaminants, particularly in sensitive areas such as coastal lagoons. However, lowering too much the concentrations of heavy metals in ethanolamine solutions can strongly diminish their antifouling power, leading to poorly effective products because of the stimulating effect exerted by the nitrogen compounds present in these solutions. Similarly, also the release of formic acid, and particularly of acetic acid, from the thermally modified wood [23] may enhance the microalgal biomass growth, as demonstrated in a fed-batch experiment with mixotrophic microalgae [66].

## 5. Conclusions

To the best of our knowledge, this is the first study investigating the effects of wood impregnating products on the development and structure of marine microalgal biofilms in a real setting such as a lagoonal system. By applying several microscopy techniques, we were able to evaluate the toxic effects of different wood treatments, i.e., copper-based coatings and thermal modification, on the microalgal abundance, community composition and diatom cell damage. Our results revealed the following:

The Cu-based agents induced modifications of the microalgal community, leading to lower densities, small-sized diatoms and frequent deformities (i.e., bent apices, frustule malformation), particularly in the genera *Cylindrotheca* and *Cocconeis.*

After 20 days, taxa forming 3D mucilaginous structures, such as *Licmophora* and *Synedra*, were present on the chemically treated panels but not on the natural ones.

While in the short term the tested treatments proved to be effective antifouling agents, in the long term they lost their effect in terms of slowing down the biofouling colonisation compared to the non-treated samples, likely due to both the leaching of contaminants and the stimulating effect of nutrients and other substances released from these solutions.

Lower hydrodynamics accelerated the colonisation, leading to higher algal biofilm abundances, regardless of the treatment.

In conclusion, to obtain a more effective antifouling effect in the long term, slightly higher concentrations of Cu (cCu > 1%) should be likely adopted to counteract the stimulating effect of the inorganic nutrients released from the antifouling solutions.

## Figures and Tables

**Figure 1 microorganisms-11-02196-f001:**
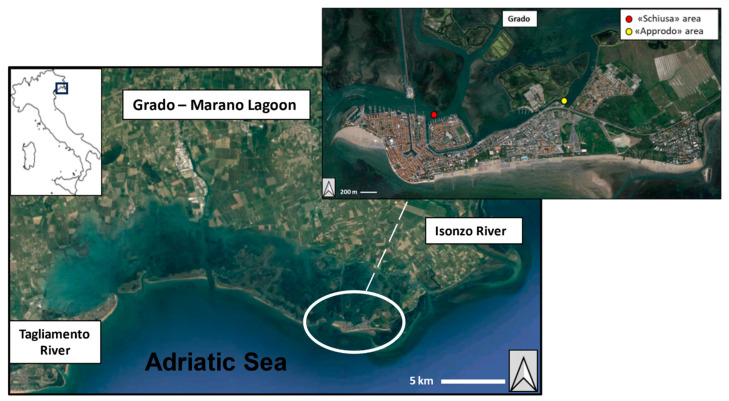
Study area in the Grado Lagoon. Detail of the two selected sites for the experiment: “Schiusa” area (red circle) and “Approdo” area (yellow circle).

**Figure 2 microorganisms-11-02196-f002:**
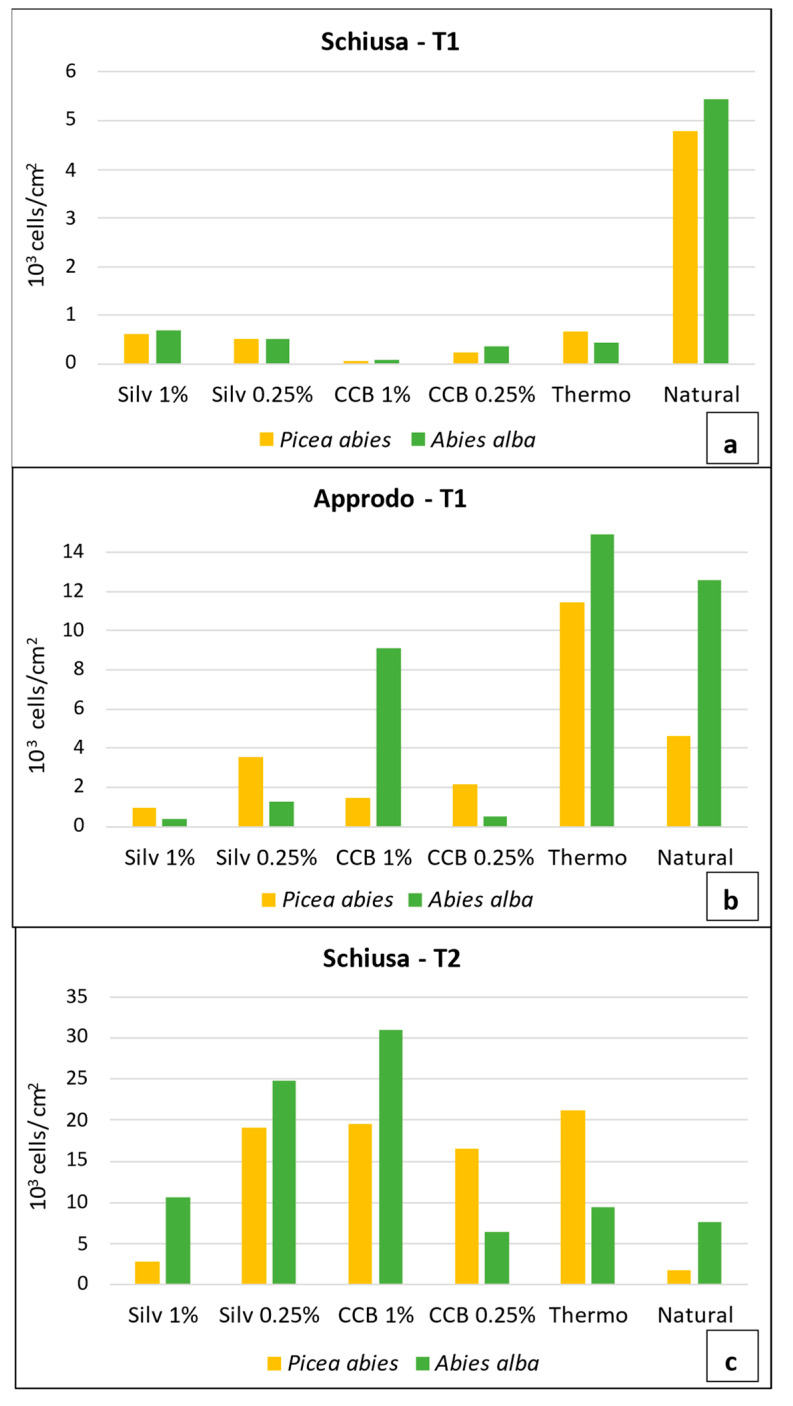
Total microalgal abundances on differently treated wooden panels in “Approdo” area (**a**) and “Schiusa” area (**b**) at T1, and in “Schiusa” area at T2 (**c**). The scales are different to better observe the pattern at each area/sampling time.

**Figure 3 microorganisms-11-02196-f003:**
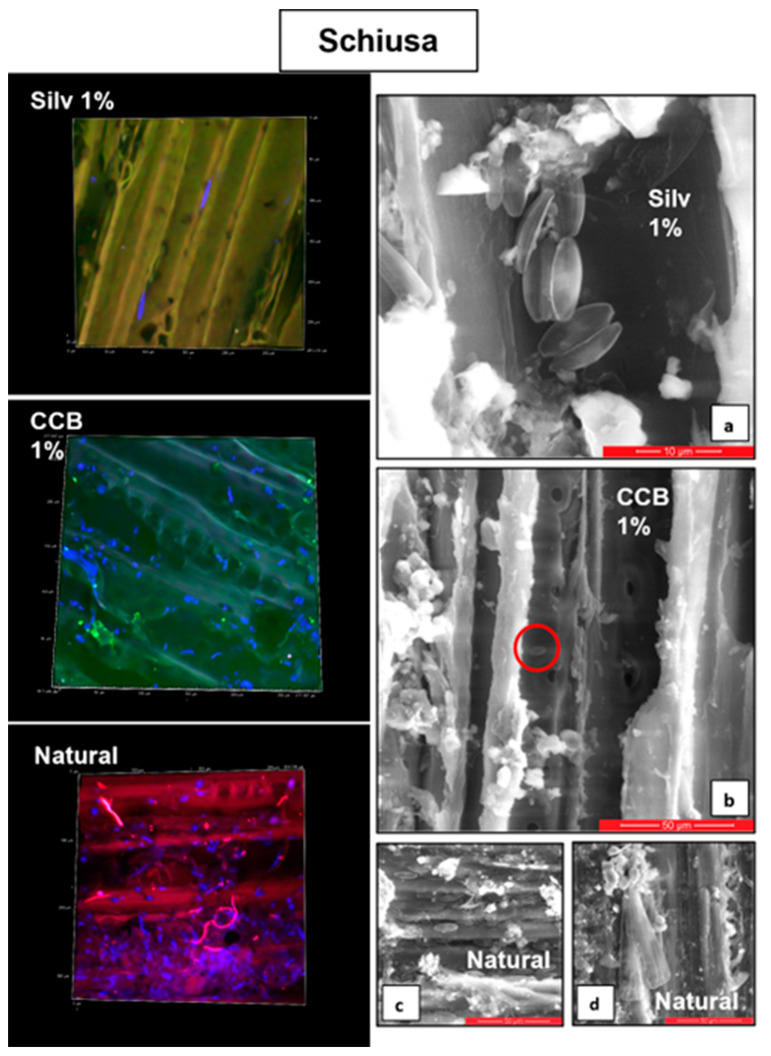
Images acquired via a laser scanning confocal microscope (LSCM, the microalgal cells have a blue, green or red colour) and scanning electron microscope (SEM) of wooden panels retrieved from the “Schiusa” area in the Grado Lagoon at T1. The LSCM images represent an area of approximately 300 µm^2^ of the wooden panel. The scale sizes in the SEM images are as follows: 10 µm (**a**) and 50 µm (**b**–**d**). The red circle in Figure (**b**) highlights the very small size of the cells in the CCB treatment.

**Figure 4 microorganisms-11-02196-f004:**
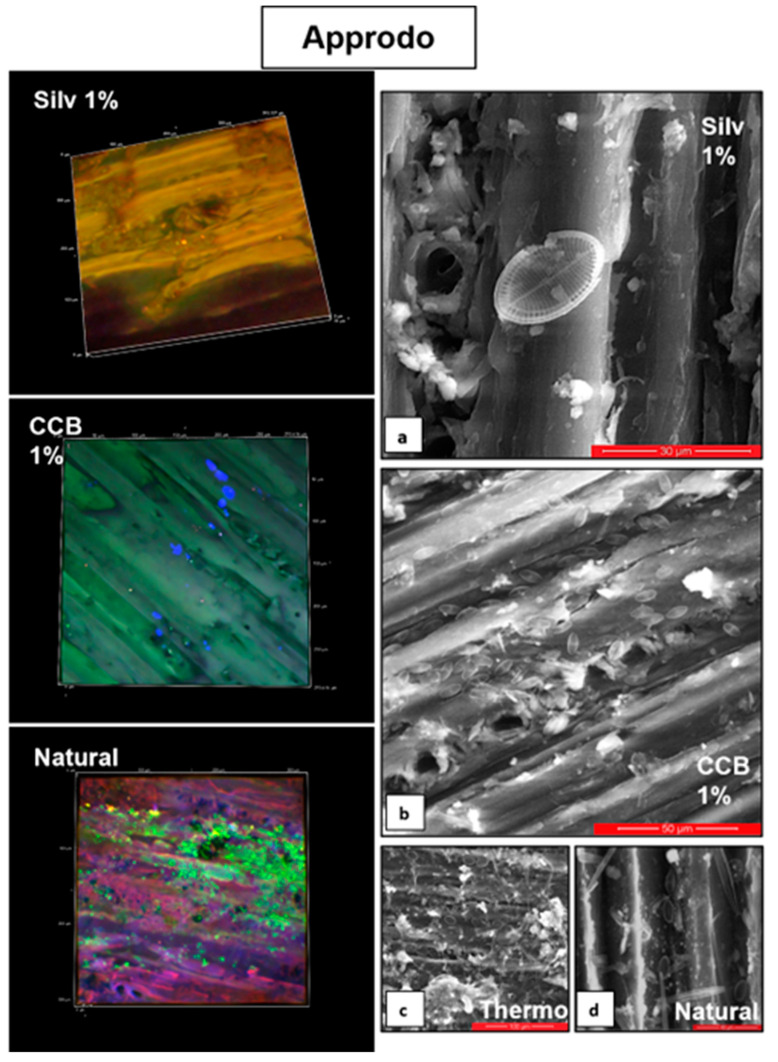
Images acquired via a laser scanning confocal microscope (LSCM, the microalgal cells have a blue, green or red colour) and scanning electron microscope (SEM) of wooden panels retrieved from the “Approdo” area in the Grado Lagoon at T1. The LSCM images represent an area of approximately 300 µm^2^ of the wooden panel. The scale sizes of the SEM images are as follows: 30 µm (**a**), 50 µm (**b**), 100 µm (**c**), and 40 µm (**d**).

**Figure 5 microorganisms-11-02196-f005:**
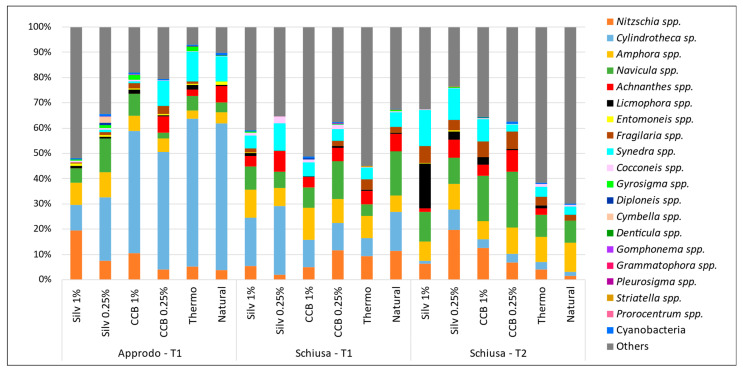
Microalgal taxonomic composition of the biofilm; relative abundances of the diatom genera and Cyanobacteria are shown. Data represent the average of the two types of wood at each sampling time. In “Others”, undetermined Pennales and Centrales are included.

**Figure 6 microorganisms-11-02196-f006:**
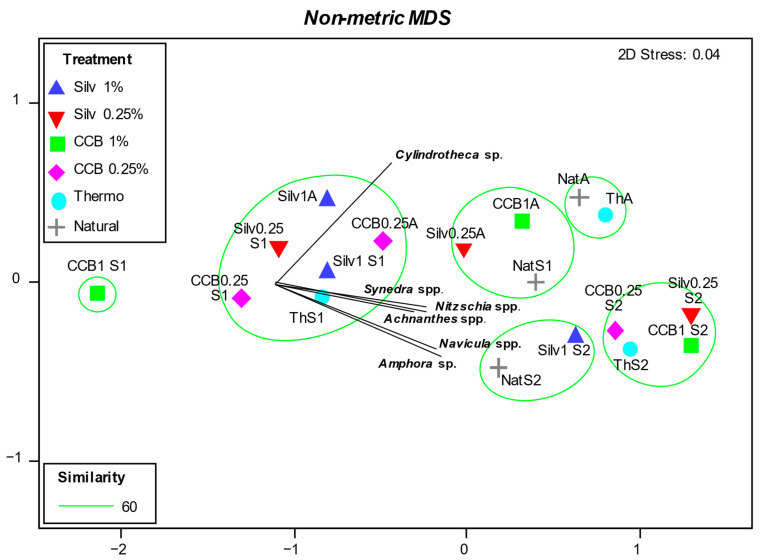
Non-metric MDS ordination plot calculated considering the total abundances of the microalgal community. The most abundant species (RA ≥ 5%) are overlaid. In the plot, the acronyms of the treatments refer to the area and the experimental time: A (“Approdo”), S1 (“Schiusa” T1), and S2 (“Schiusa” T2). Silv = Silvanolin; Thermo or Th = thermally modified.

## Data Availability

The data presented in this study are available on request from the corresponding author.

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
