# Peer review of "Ecological Effect of Differently Treated Wooden Materials on Microalgal Biofilm Formation in the Grado Lagoon (Northern Adriatic Sea)"

_microorganisms, 2023, doi:10.3390/microorganisms11092196_

Round 1

Reviewer 1 Report

In this manuscript, the authors analyzed the impact of different wood treatments on the formation of microbial biofilm by several microscope techniques in the Grado Lagoon. Through the changes in microalgae abundance, community composition and diatom cell damage, the toxic effects on the environment under different treatment conditions were evaluated. This manuscript reported that in the short term, the tested treatments were effective antifouling agents, but in the long term, it was necessary to slightly increase the concentration of copper to achieve the effective antifouling effect. Therefore, I suggest it be accepted after the author complements the content, articulates the obscure content, and corrects the format.

1.      The abstract did not completely summarize the content of the article, and the influence of copper-based coating treatment was summarized but lacks the influence of thermal modification treatment on microorganisms.

2.      It is suggested to add a conclusion section at the end of the article, mainly summarizing the research results of this article.

3.      The generic names of bacteria should be italicized throughout the article, such as “Cylindrotheca” etc.

4.      Unit writings were not standardized, such as the “km2” in line 123 and “cm2” in line 225.

5.      There was no need to add the abbreviation of dissolved oxygen in line 148, which did not appear in the following text.

6.      Subscript writings on chemical molecular formulas were not standardized, for example, the “CaMg(CO3)2” should be changed to “CaMg(CO3)2” in line 213.

7.      The legend in Figure 1 was not consistent with that in the article, with “Schiusa” in the article and《Schiusa》. And there is no “《》” symbol in English.

8.      Suggested that mark the source of the formula in part 2.3.

9.      The use of spaces. There seems to be something wrong with the space between “N*” and “VF” in the formula in 2.3 and no spaces are needed before and after “±”.

10.   The fonts of ANOVA symbols were not italicized, such as the “p” in line 272.

11.   The fonts in the picture were not uniformly in Times New Roman font. And axis and scale lines should be added to the figure 2 and figure 5.

12.   The figure note format was not uniform. “Figure 2” is not bold, while others are.

The language can be improved further. 

Reviewer 2 Report

The manuscript entitled “Ecological effect of differently treated wooden materials on microalgal biofilm formation in the Grado Lagoon (northern Adriatic Sea)” is devoted to on the impact of different wood treatments (copper-based coatings and thermal modification) on microbial biofilm formation in in marine or lagoon environments (Grado Lagoon). Preserving wooden structures in sea water is urgent problem for wood infrastructures such as piers, moorings, piling and fences which are part of the local heritage, so to my mind this manuscript. is topical and corresponding to the aims and scopes of the Microorganisms journal.

After reading the manuscript I have several comments.

1.     The introduction should add the role of copper-containing wood treatments to prevent fouling by microbial biofilms, which can be pioneer species, creating a defense effect for microalgae and macrofouling. In my opinion, this is an important issue in the development of materials that prevent multispecies biofouling.

2.     at the beginning of the results it is necessary to characterize the obtained materials, first of all by real copper concentration, it is desirable to compare materials prepared by different methods by surface area, which is important for biofouling.

3.     I wanted to see the reason for the differences in biofouling of materials from the two pine species.

4.     L 357 By analysing the composition of the community at the genus level,

There's no mention of bioinformatics analysis in the materials and methods. Add a full description of the methodology, DNA extraction, amplification, sequencing, etc.

5.     On the basis of bioinformatic analysis, it is worthwhile to define diversity indices that can be used for biofouling assessment. This may provide important data for discussion.

6.     In addition to the seawater test, data on copper yields from materials under laboratory conditions should be provided. The authors state that the copper in the materials is not toxic to other inhabitants. This needs to be demonstrated.

7.     The discussion speculates on the mechanisms of copper toxicity to algae, but the paper does not assess this. It is worth paying less attention to these literature facts.

8.     Most importantly, the manuscript lacks a conclusion in which the authors should show how their method can reduce biofouling of already existing wooden objects.

Minor editing of English language required

Round 2

Reviewer 2 Report

having carefully read the revised manuscript, I can say that my remarks have been taken into account. I believe that the text can be published in this form.